# Expression of c-MET in Estrogen Receptor Positive and HER2 Negative Resected Breast Cancer Correlated with a Poor Prognosis

**DOI:** 10.3390/jcm11236987

**Published:** 2022-11-26

**Authors:** Francesco Iovino, Anna Diana, Francesca Carlino, Franca Ferraraccio, Giuliano Antoniol, Francesca Fisone, Alessandra Perrone, Federica Zito Marino, Iacopo Panarese, Madhura S. Tathode, Michele Caraglia, Gianluca Gatta, Roberto Ruggiero, Simona Parisi, Ferdinando De Vita, Fortunato Ciardiello, Ludovico Docimo, Michele Orditura

**Affiliations:** 1Department of Translational Medical Science, School of Medicine, University of Campania Luigi Vanvitelli, 80131 Naples, Italy; 2Department of Precision Medicine, School of Medicine, University of Campania Luigi Vanvitelli, 80131 Naples, Italy; 3Department of Mental and Physical Health and Preventive Medicine, School of Medicine, University of Campania Luigi Vanvitelli, 80131 Naples, Italy; 4Départment de Génie Informatique et Génie Logiciel-Polytechnique Montréal, 2500, Chemin de Polytechnique Montreal, Montreal, QC H3T 1J4, Canada; 5Department of Advanced Medical and Surgical Sciences, School of Medicine, University of Campania Luigi Vanvitelli, 80131 Naples, Italy

**Keywords:** biomarkers, early breast cancer, hepatocyte growth factor receptor, c-MET

## Abstract

Introduction: The mesenchymal-epithelial transition factor (c-MET) receptor is overexpressed in about 14–54% of invasive breast cancers, but its prognostic value in clinical practice is still unclear. Methods: In order to investigate the relationship between c-MET expression levels and prognosis, we retrospectively reviewed the clinical features and outcomes of 105 women with estrogen receptor positive HER2 negative (ER+/HER2-) resected breast cancer. We used the Kaplan Meier method to estimate Disease Free Survival (DFS) and Breast Cancer Specific Survival (BCSS) in the subgroups of patients with high (≥50%) and low (<50%) c-MET expression. Univariate and multivariate Cox proportional regression models were performed to assess the prognostic impact of clinicopathological parameters for DFS an BCSS. Results: High c-MET values significantly correlated with tumor size, high Ki67 and low (<20%) progesterone receptor expression. At a median follow up of 60 months, patients with high c-MET tumor had significantly worse (*p* = 0.00026) and BCSS (*p* = 0.0013). Univariate analysis showed a significant association between large tumor size, elevated Ki67, c-MET values and increased risk of recurrence or death. The multivariate COX regression model showed that tumor size and high c-MET expression were independent predictors of DFS (*p* = 0.019 and *p* = 0.022). Moreover, large tumor size was associated with significantly higher risk of cancer related death at multivariate analysis (*p* = 0.017), while a trend towards a poorer survival was registered in the high c-MET levels cohort (*p* = 0.084). Conclusions: In our series, high c-MET expression correlated with poor survival outcomes. Further studies are warranted to validate the clinical relevance and applicability of c-MET as a prognostic factor in ER+/HER2- early BC.

## 1. Introduction

Breast cancer (BC) is a widely heterogeneous disease which translates to different clinical approaches and treatment sensitivity [1]. Recently, several gene expression assays have been endorsed by major international guidelines as prognostic tools in early BC (stage I, II and node negative) [2]. However, these multigene prognostic tests are not widely available for routine clinical practice, therefore therapeutic decisions are still based on traditional clinicopathological parameters such as histological subtype, grading, tumor stage, hormonal receptors expression, human epidermal growth factor receptor (HER2) status and proliferation index. During recent decades, several efforts have been spent to identify specific molecular alterations in order to facilitate the development of novel targeted therapies [3].

Dysregulation of hepatocyte growth factor/mesenchymal-epithelial transition factor gene (HGF/c-MET) pathway plays a crucial role in human carcinogenesis, tumor progression and resistance to antineoplastic treatments. The activation of the HGF/c-MET axis is responsible for the recruitment of four signalling cascades: mitogen-activated protein kinase/extracellular signal-regulated kinase (MAPK/ERK), Phosphatidylinositol 3-kinase (PI3K/AKT), Nuclear factor-kappa B (NF-κB), and Janus kinase/signal transducer and activator of transcription (JAK/STAT) 3/5. Once activated, these signalling pathways promote cell proliferation, invasion, angiogenesis and epithelial to mesenchymal transition (EMT).

The abnormal activation of c-MET axis is triggered by various molecular mechanisms, including paracrine signalling, autocrine loop formation, germline or somatic mutations, chromosomal rearrangement, *c-MET* amplification, and c-MET protein or HGF overexpression [4]. Genetic alterations of *c-MET* proto-oncogene are extremely rare, while *c-MET* amplification or c-MET receptor overexpression have been reported in several tumors, such as lung, pharyngeal, gastric and breast cancers [5]. c-MET protein, a tyrosine kinase (TK) receptor that binds HGF, is overexpressed in about 14–54% of invasive breast cancers [6]. In preclinical models, c-MET expression levels were higher in poorly differentiated and invasive cancer cell lines [7]. Additionally, a large body of evidence showed a strong correlation between high level of c-MET expression and Basal-like subtype [8,9,10].

During the last few years, several trials and metanalyses have explored the role of c-MET deregulation as prognostic factor in different cancers, suggesting a correlation with worse survival outcomes [11]. Clinical trials that were focused on the prognostic significance of c-MET overexpression in BC and its correlation with specific BC subtypes reported controversial results, thus the clinical utility of c-MET evaluation is still unconfirmed [6,10,12,13].

We retrospectively evaluated the c-MET expression value in a cohort of early Estrogen Receptor positive, HER2 negative (ER+/HER2-) resected BC patients to analyze the relationship between c-MET levels and survival outcomes.

## 2. Materials and Methods

### 2.1. Study Population

The patients included in the present study were diagnosed and treated at the Surgery and Oncology Departments of the University of Campania “L. Vanvitelli”, Naples, Italy, according to the established guidelines for clinical practice.

The expression of c-MET was assessed in a total of 105 surgical samples from patients with histologically proven ER+/HER2- (T1-T3, N0-2) BC treated at our Institution from July 2009 to January 2018.

We excluded 50 cases of breast carcinoma in situ, 60 locally advanced and 30 metastatic BC at the time of diagnosis and 30 patients who received neo-adjuvant chemotherapy. We collected the following clinicopathological data: age, sex, menopausal status, histological type, tumor stage, grade of differentiation, hormonal receptor and HER2 expression status, Ki67 index, and type of adjuvant treatment. The clinicopathological characteristics are summarized in Table 1. The study was carried out in accordance with relevant guidelines/regulations and with the ethical principles of the Declaration of Helsinki. In compliance with patients’ rights, participants’ identities were kept confidential. All the patients at the moment of their hospitalization provided informed consent for using their biological and histological samples. Therefore, the ethical committee approved the study of the University of Campania “L. Vanvitelli” (prot n. 0017554/2022).

### 2.2. Pathological Evaluation

Tumors were considered ER positive if 1% or more of tumor cells demonstrated positive nuclear staining on immunohistochemistry. We set a cut-off point to distinguish low versus high Ki67 expression at 20%. Progesterone receptor (PR) expression was considered high in the presence of nuclear staining in 20% or more cells, according to the 2013 St. Gallen International Breast Cancer Conference. Fluorescent in situ hybridization (FISH) test for HER2 was performed in case of equivocal results on IHC (2+).

### 2.3. Immunohistochemistry Analysis

Immunohistochemistry (IHC) analysis was performed on the automated Ventana Benchmark Ultra with ultraView Universal DAB Detection Kit (Ventana Medical Systems, Tucson, AZ, USA) using a previously described protocol [14]. IHC test was performed on FFPE sections using the commercially available antibodies anti-c-MET SP-44 ready to use Rabbit Monoclonal Primary Antibody (Ventana, Catalog Number: 790-4430); anti-(ER) (SP1) Rabbit Monoclonal Primary Antibody ready to use (Ventana, Catalog Number: 790-4325); anti-Progesterone Receptor (PR) (1E2) Rabbit Monoclonal Primary Antibody ready to use (Ventana, Catalog Number: 790-2223); anti-HER-2/neu (4B5) Rabbit Monoclonal Primary Antibody ready to use (Ventana, Catalog Number:790-100); anti-Ki-67 (30-9) Rabbit Monoclonal Primary Antibody ready to use (Ventana, Catalog Number: 790-4286). All antibodies were incubated for 4 min at a temperature of 37 °C.

We used human tonsil tissue as positive control to evaluate the low expression in macrophages in germinal centers and the high expression in the crypt epithelium (provided by Novus Biologicals, Biotechne, Minneapolis, MN, USA). IHC results were interpreted as follows: low positivity: MET immunostaining in 1℃49% of neoplastic cells; high positivity: MET immunostaining in more than 50% of neoplastic cells [15]. (Figure 1). The expression of c-Met was also evaluated in normal adjacent tissue. The surgical samples collected from patients and shown in Figure 2 corresponded to tumor tissues and adjacent non-tumor tissues.

### 2.4. Fluorescence In Situ Hybridization (FISH)

The FISH test was performed using the Bond FISH kit (Catalog Number: DS9636) on the automated Bond system (Leica Biosystems) using a previously described protocol [16]. The FISH test was performed on FFPE whole sections using the commercially available Kreatech FISH probe and MET (7q31)/SE 7 (D7Z1)-XL for Bond (Catalog Number: KBI-XL003). The combined probes are used to detect amplification of the MET gene at 7q31, with the centromere probe as a control. The MET (7q31)—XL probe was optimized to detect copy numbers of the MET gene region at 7q31. The SE7 (D7Z1)—XL probe was optimized to detect copy numbers of the centromere region of chromosome 7. The analysis was performed in at least 100 nuclei. FISH test results were evaluated as follows: MET amplification: MET/CEP7ratio ≥ 2.0 or mean of the signals of MET/cell ≥ 6; MET gene copy number: mean of the signals of MET/cell ≥4 to <6; MET negative: MET/CEP7 ratio < 2.0 and MET-CN < 4.

### 2.5. Statistical Analysis

We collected and reported continuous variables (e.g., ER and Ki67), discrete variables (e.g., age) and categorical variables (e.g., grading, T). In order to ease both analysis and interpretation, we dichotomized nine additional variables, quantized into two groups according to the following thresholds: age (<50 vs. ≥50), c-MET (<50% vs. ≥50%), Ki67 (<20% vs. ≥20%), PR (<20% vs. ≥20%), Grade (G1-G2 vs. G3), histological type (NST vs. other), lymph nodal status (N0 vs. N1-N2), tumor size (T1 vs. T2-T3), pathological tumor stage (stage I-II vs. III).

We compared clinicopathological parameters according to c-MET protein expression status using Chi-square test. We used Pearson’s Chi-square test to determine whether there was a statistically significant interdependence between the categorical variables. In order to verify the presence of an association between low/high c-MET values and other variables, we also applied the unpaired two samples Wilcoxon test, also known as the Wilcoxon rank sum test or Mann-Whitney U test. Finally, to compare IHC staining levels, we applied the paired Wilcoxon signed rank test.

Disease free survival (DFS) was measured from the date of diagnosis to the date of first relapse or the last follow up date without evidence of disease progression. Breast cancer specific survival (BCSS) was defined as the time interval between diagnosis and the last follow up or cancer death.

To model BCSS and DFS, we applied the Cox proportional hazard regression model. To select the subset of variables to be retained for multivariate analysis, we applied a univariate hazard model to each predictor variable. Clinicopathological variables which resulted as statistically significant under univariate analysis were included in the Cox multivariate analysis to assess the prognostic value of these factors. In all statistical tests, we fixed significance level, the probability of rejecting the null hypothesis when it is true, at 0.05. Survival rates of the two subgroups, defined according to c-MET expression level (high versus low), were assessed using Kaplan Meier curves and compared by log rank test.

All statistical analyses were performed with the R environment, version 3.6.2, on a 64 bits MacBook Pro.

### 2.6. Pathway and Gene Ontology Enrichment Analysis

Pathway and gene ontology enrichment analysis was performed using online g:Profiler [https://biit.cs.ut.ee/gprofiler/gost] accessed on 14 November 2022, with adjusted *p*-value threshold of 0.05. Pathway databases used for the analysis included KEGG, Reactome and WikiPathways. Enrichment analysis was performed using c-Met [ENSG00000105976] as input gene along with Estrogen Receptor 1 [ENSG00000091831], Estrogen Receptor 2 [ENSG00000140009], Progesterone Receptor [ENSG00000082175] and Erb-B2 Receptor Tyrosine Kinase 2 [ENSG00000141736].

### 2.7. GEPIA2 Survival Analysis

Clinical significance of c-Met in clinical Breast Cancer patients (TCGA-BRCA) was evaluated using Kaplan–Meier plot analysis. This analysis was performed using GEPIA2 (Gene Expression Profiling Interactive Analysis) online tool (http://gepia2.cancer-pku.cn/#index) accessed on 11 November 2022, using RNASeq expression datasets from The Cancer Genome Atlas- Breast Cancer (TCGA) and the Genotype-Tissue Expression (GTEx). Overall survival (OS) and Disease-free Survival (DFS) plots were derived using “Survival analysis” module of GEPIA2 with parameters including 95% confidence level, “Media” group cut-off and Hazard Ratio with cohort size for low and high expression of C-Met was 534 and 535, respectively.

### 2.8. Human Protein Atlas (HPA) Analysis

Human Protein Atlas (HPA) is comprehensive and publicly available database which contains spatial distribution of proteins in human tissues and cells derived using immunohistochemistry and immunocytochemistry, respectively. Protein expression pattern of c-MET was studied using HPA to understand its clinical significance in breast cancer.

## 3. Results

### 3.1. Patient Characteristics

We enrolled 105 patients with breast cancer, mean age 60 years with a median of 59 and a range of 49–72. Of the 105 patients, 30 were younger than 50 years. Sixty-six patients were T1 and 39 T2-T3, 65 patients had no lymph node metastases and 40 N1-N2, with 46 patients with Stage I, 39 Stage II and 20 Stage III disease, respectively. High-intermediate grade of differentiation was recorded in 83 patients and low grade of differentiation in 22 patients; 45 tumours had low and 60 high proliferation markers, respectively; perineural invasion was absent in 87 tumours and present in the 18 remaining cases. We recorded NST in 90, lobular in 11 and other histo-types in four patients, respectively. Chemotherapy was performed in 60 patients (see Table 1). 

Interestingly, no expression of c-Met was recorded in the normal tissue except in four samples, where the expression was low, whereas only in five cancer tissues was c-Met not expressed. Indeed, the c-Met percentage distribution in the studied population largely differs between normal and cancer tissues (see the boxplot Figure 2).

Therefore, on the presence/absence of c-Met (quantized values) in normal/cancer tissue, the Chi-square test, Table 2, rejected (with a *p*-value <0.001) the null hypothesis: rows/columns were independent. Indeed, the Fisher exact test reported an odds ratio of 453 (with a *p*-value <0.001). The differences in the expression of low/high c-MET in normal and cancer tissue counterparts were highly statistically significant. The results were also confirmed by the staining percentage levels of c-Met (non-quantized—see Figure 2); the paired samples Wilcoxon rank-signed test, two-sided, between the c-Met percentage (cancer tissue versus normal tissue) rejected with very strong statistical evidence, *p*-value < 0.0001, the null hypothesis that there was no difference in cancer/normal tissues’ c-Met distribution.

### 3.2. Correlation between High c-MET and Clinicopathological Variables

Among the 105 patients, 43 (41%) had high c-MET expression in the tumor specimen while amplification of c-MET gene was not detected in any of the cases (Figure 3).

A significant correlation between high c-MET and tumors ≥2 cm (56% vs. 44%, *p* = 0.023), high Ki67 values (58% vs. 42%, *p* < 0.001) and low PR expression status (63% vs. 37%, *p* = 0.013) was observed. Conversely, the majority of stage I BC showed low c-MET expression (74% vs. 26%, *p* = 0.023).

All patients received adjuvant endocrine therapy whereas post-operative chemotherapy was administered in case of luminal B-like phenotype with higher risk of recurrence (high grade, large tumor size, nodal involvement), according to international guidelines. Overall, 43% of the study population did not receive adjuvant chemotherapy, two-thirds of whom had low c-MET levels (76% vs. 24%, *p* = 0.05). No significant correlation with age, nodal involvement, tumor grade and histological subtype was reported (Table 3).

### 3.3. Survival Analysis

At a median follow up of 60 months, disease recurrence and cancer-related deaths were registered in 17 (16%) and 16 (15%) patients, respectively. The c-MET overexpression was significantly related to worse DFS (*p* = 0.00026) and BCSS (*p* = 0.0013) (Figure 4 and Figure 5).

At univariate analysis, a significant higher risk of recurrence or death was registered for patients affected by high Ki67, elevated c-MET and stage III tumors (Table 3).

The multivariate regression analysis confirmed tumor stage and c-MET expression as independent prognostic factor for both DFS and BCSS. Specifically, women with stage III tumors or high c-MET expression had three to eight times higher risk of relapse than those affected by early stage (I-II) or low c-MET BC, respectively (HR: 3.15, 95% CI: 1.18–8.18, *p* = 0.021 and HR: 8.58, 95% CI: 1.95–37.79, *p* = 0.004).

Similarly, this subset of patients had also a significant increased risk of cancer death (HR: 3.071, 95% CI: 1.10–8.56, *p* = 0.032 and HR: 11.09, 95% CI: 1.44–85.35, *p* = 0.021).

### 3.4. Pathway and Gene Ontology Enrichment Analysis

List of pathways and gene ontology terms enriched by c-MET along with Estrogen Receptor 1, Estrogen Receptor 2, Progesterone Receptor, Erb-B2 Receptor Tyrosine Kinase 2 together is described in Appendix A. Analysis shows involvement of input genes considered altogether in molecular functions which include signalling receptor activity, molecular transducer activity and enzyme binding. Moreover, the list of enriched pathways includes generic transcription pathway, RNA polymerase II transcription and gene expression. A detailed list of all pathways and gene ontology terms is provided in Appendix A.

### 3.5. GEPIA2 Survival Analysis

As per GEPIA2 survival analysis using existing clinical datasets from TCGA and GTEx, increased expression of c-Met RNA in BRCA clinical patients is associated with poor OS, even if this difference did not reach statistical significance (Figure 6). This in-silico analysis partially supports our experimental findings from IHC in ER+/HER2- resected breast cancer patients. In fact, this analysis correlates the expression of c-Met RNA and not of the Met protein as performed in our experimental model using immunohistochemical evaluation on paraffin-embedded BRCA tissues. Moreover, no information on the eventual correlation in the specific ER+/HER2- resected breast cancer patients can be derived from the currently available data set. On these bases, the diverse correlation observed in the GEPIA2 survival analysis could, therefore, be based on those pitfalls.

### 3.6. Human Protein Atlas (HPA) Analysis

As per protein expression associated pathological data in HPA, we found high/medium protein level expression of c-MET in breast tumor tissues (Appendix A). Our experimental observation falls under the same pattern of c-Met protein expression, particularly in ER+/HER- Breast cancer tumor tissues.

## 4. Discussion

c-MET is a tyrosine kinase receptor mainly expressed by epithelial cells in physiological conditions [17,18]. An over-activation of c-MET pathway plays a key role in carcinogenesis, tumor progression and resistance to antineoplastic treatments in various human cancer types [11,19].

Even though several trials reported a correlation between c-MET overexpression and worse clinical outcomes, its prognostic value remains controversial in BC [6,20]. Elevated levels of c-MET protein are responsible for most cases of c-MET axis deregulation in BC, while mutation or amplification of *c-MET* encoding gene are extremely rare, suggesting that c-MET overexpression is induced at the transcriptional level [9,21]. In our research, tumors with ≥50% of neoplastic cells of intensity immunostaining were defined as high c-MET and were diagnosed in 41% of the study population. Increased c-MET expression has been reported in almost 14–54% of BC patients, and is more likely found in tubular histology, TNBC and, above all, in basal-like molecular subtype [6,9]. This wide range of c-MET expression is mainly related to the use of several antibodies selected for immunohistochemical analysis and different definitions of high c-MET expression; therefore, a standardized method of c-MET evaluation should be established. Due to the low evidence from previous studies, our research was aimed to assess the prognostic significance and the potential clinical implication of high c-MET expression among women with ER+/HER2-BC. In line with the results from previous retrospective studies and meta-analysis [22,23,24,25,26,27,28], in our cohort patients affected by high c-MET or stage III tumors had a statistically significant worse survival as both DFS and BCSS.

Notably, a comprehensive meta-analysis involving 6010 BC patients revealed that c-MET overexpression was associated with 1.41-fold increased risk of recurrence in ER+/HER2- resected BC patients, likely due to the induction of EMT gene expression signature supporting cancer metastases [22]. However, the remarkable differences both in the study population and in the c-MET expression evaluation method do not allow for a direct comparison between our study and previous trials [22,23,24,25,26,27,28]. Recently, several gene expression assays have been endorsed by major international guidelines as prognostic tools in ER+/HER2-early BC [2]. These multigene tests are not widely available for routine clinical practice, and the therapeutic decisions are still based on traditional clinicopathological parameters. In this scenario, our findings support the assessment of c-MET expression, in addition to the conventional clinico-pathological features, for risk prediction in ER+/HER2- resected BC patients by using simple IHC analysis. The use of this assay allows a low cost and easy technical procedure that provides a strong predictor of clinical evolution of BC in a subset of patients for which it is essential to decide the therapeutic adjuvant strategy in order to prevent relapse. It is noteworthy that the addition of molecular parameters in the definition of a tumour disease allows the development of treatment modalities based upon precision medicine specifically treating the single patient.

Moreover, in contrast to some prior studies where no association with clinico-pathological characteristics or mixed results were found [20,28,29], our findings showed a statistically significant correlation between c-MET overexpression and tumor size and stage, PR levels and Ki67 index.

On the other hand, our study has the following limitations: (i) it is a retrospective study with a limited number of cases; (ii) there is currently no widely accepted standardized methodology for immunohistochemical staining and scoring of c-Met generating bias of data interpretation.

Although these data do not allow definitive conclusions, they demonstrate a crucial role of the HGF/c-MET pathway in tumor cell proliferation and clinical behavior and highlight the need for additional investigations on its molecular mechanism in order to promote both the development of anti-MET targeted therapies and the generation of a new prognostic score and decisional algorithm in BC.

During recent decades, several efforts have been made to identify specific molecular alterations in order to facilitate the development of novel targeted therapies [3]. In the case of c-Met in BC, it has been suggested that HGF/MET signalling is a mediator of resistance to anti-cancer immunotherapy, following the rationale of using anti-MET drugs in combination with immunomodulatory agents [30]. On this basis, it is an attractive therapeutic strategy to target BC cells overexpressing c-Met through immunological approaches based upon CAR-T engineered to specifically recognize c-MET overexpressing BC cells. The last strategy can have an important therapeutic effect on the generation of a strong inflammatory anti-cancer response in the tumour tissue, which can also target a c-MET non-expressing cancer cells population. In this way, it should be possible to overcome immune resistance of c-Met overexpressing BC cells. Other therapeutic strategies (TK inhibitors and monoclonal antibodies) targeting HGF/c-MET signalling pathways are currently under evaluation in metastatic BC, especially in BL or TNBC settings. In this light, the crosstalk between MET and ER pathways, confirmed also in our study, has pushed the development of c-MET inhibitors in combination with endocrine agents as an attractive therapeutic strategy to delay and overcome endocrine resistance [31,32].

Overall, despite the small sample size, the lack of a standardized evaluation method and cut-off and the limitations of a retrospective study, our findings suggest the validity of c-MET evaluation for risk stratification process encouraging future clinical trials involving a large population of carefully selected patients to better elucidate its prognostic impact and to evaluate the efficacy of MET inhibitors in luminal subtype [33].

## Figures and Tables

**Figure 1 jcm-11-06987-f001:**
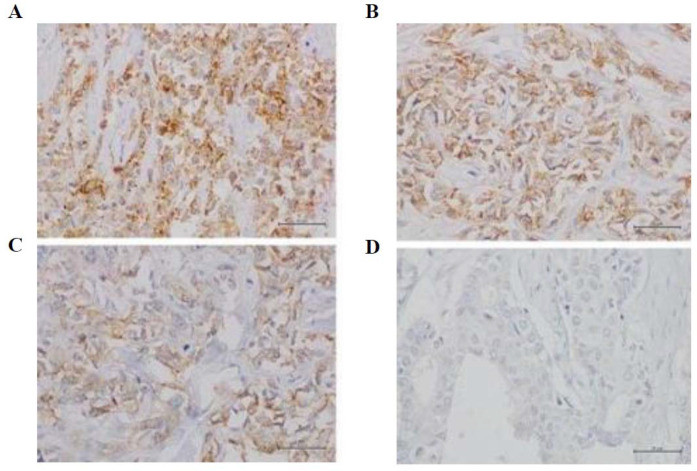
Immunohistochemical stain for c-MET receptor; (**A**) score 76–100%; (**B**) score 51–75%; (**C**) score 26–50%; (**D**) score 0–25%. 40× magnification.

**Figure 2 jcm-11-06987-f002:**
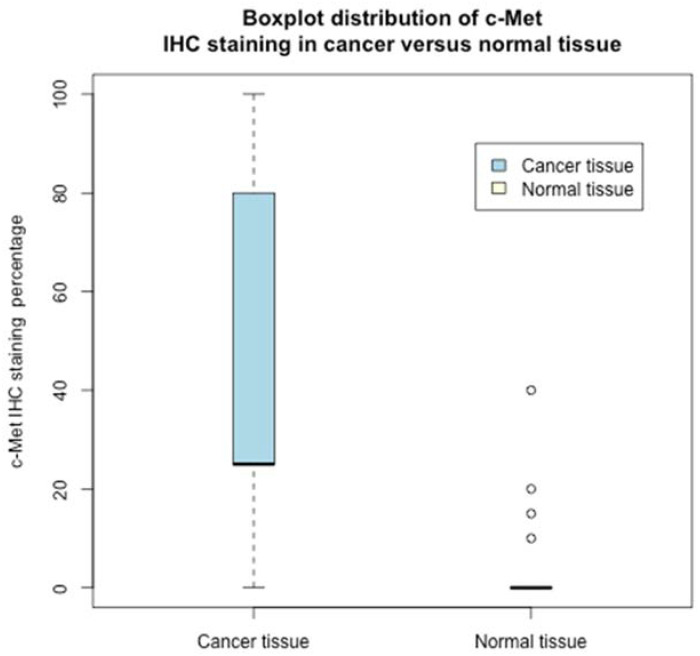
Boxplot distribution of c-Met IHC staining in cancer versus normal tissue.

**Figure 3 jcm-11-06987-f003:**
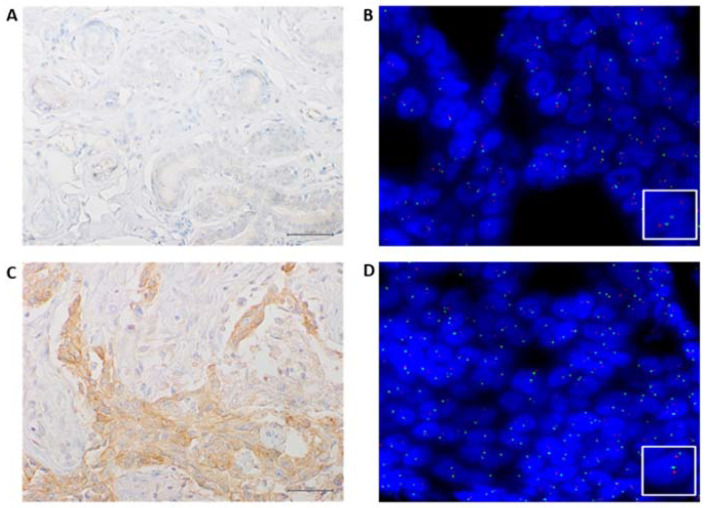
Representative results of MET IHC and FISH of a case analyzed in our series; (**A**) negative immunostaining of C-MET in a normal area (40× magnification); (**B**) MET FISH not amplified in a normal area, as indicated by two orange signals (MET) and two green (SE7) signals in each nucleus (60× magnification); (**C**) positive immunostaining of C-MET in a tumoral area (40× magnification); (**D**) MET FISH not amplified in a tumoral area as indicated by two orange signals (MET) and two green (SE7) signals in each nucleus (60× magnification).

**Figure 4 jcm-11-06987-f004:**
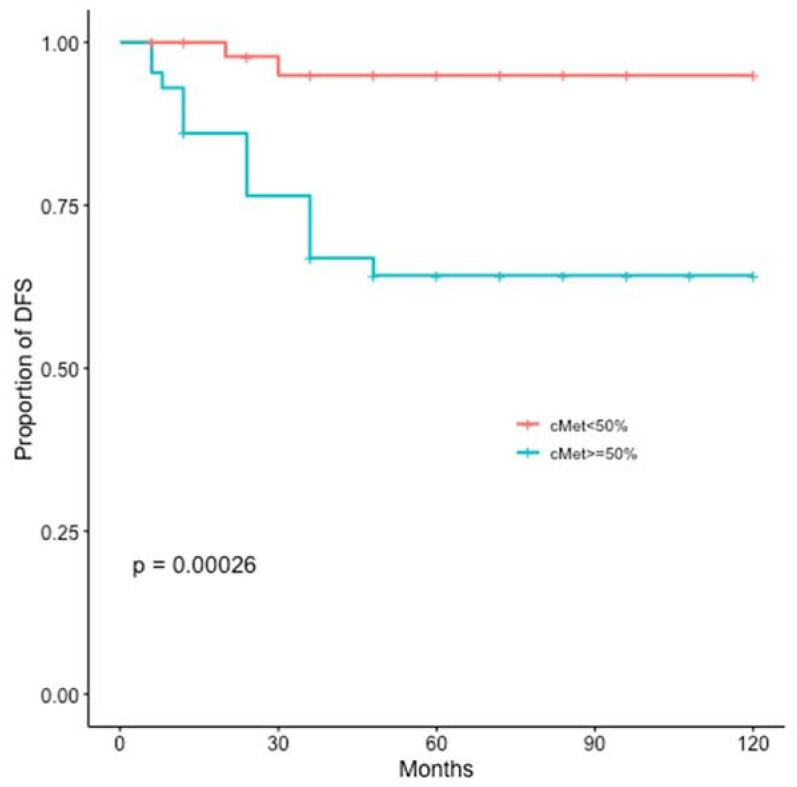
Kaplan-Meier estimates illustrating the DFS of patients by high c-MET (≥50%) and low c-MET (<50%) expression levels.

**Figure 5 jcm-11-06987-f005:**
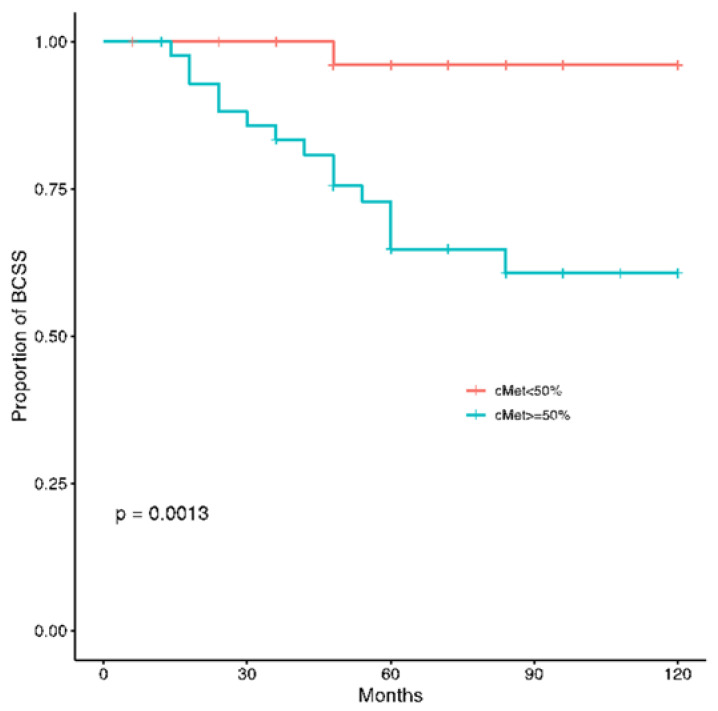
Kaplan-Meier estimates illustrating the BCSS of patients by high c-MET (≥50%) and low c-MET (<50%) expression levels.

**Figure 6 jcm-11-06987-f006:**
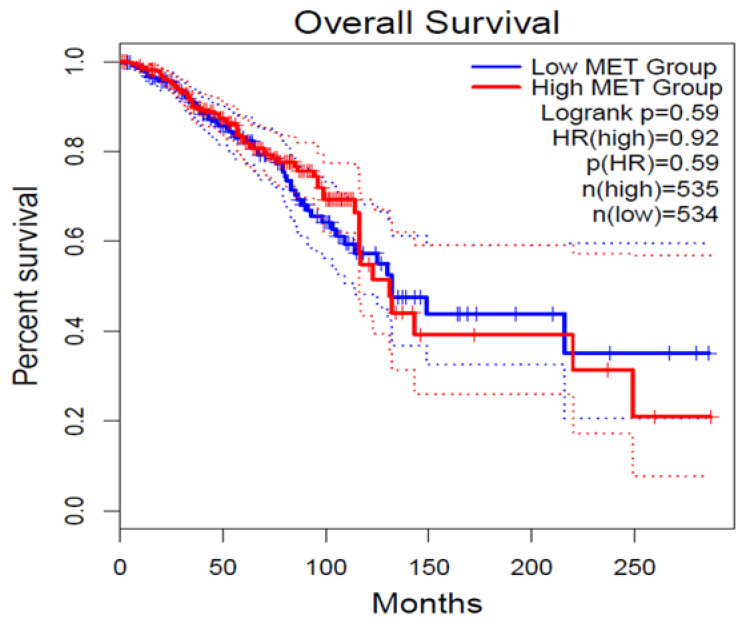
Overall Survival (OS) plot for c-MET expression derived from online GEPIA2 tool for set of high (red) and low (blue) c-MET expression level cohort. ‘N’ represents the size of the patient cohort involved in the study.

**Table 1 jcm-11-06987-t001:** Patient characteristics.

Cohort	
Patients	105
Age	
median(range)	59 (49–72)
mean (standard deviation)	60 (13.52)
<50 y	30
≥50 y	75
Tumor size	
T1	66
T2	30
T3	9
Lymph nodal status	
N0	65
N1	23
N2	17
Tumor stage	
Stage I	46
Stage II	39
Stage III	20
Grade	
G1	9
G2	74
G3	22
Ki67	
<20%	45
≥20%	60
Perineural invasion	
Absent	87
Present	18
Hystological type	
NST	90
Lobular	11
Other	4
ER	
Negative	0
Positive	105
PR	
Negative	27
Positive	78
HER2	
Negative	91
Positive	14
c-Met	
<50%	62
≥ 50%	43
Hormonotherapy	
Yes	97
No	8
Chemotherapy	
Yes	60
No	45

No special type, NST; estrogen receptor, ER; progesterone receptor, PR.

**Table 2 jcm-11-06987-t002:** Associations between c-MET expression level (high c-MET (≥50%) and low c-MET (<50%)), and clinicopathological characteristics of patients.

Characteristics	N (%)	High c-MET (N, %)	Low c-MET (N, %)	Χ^2^	*p*-Value
Total	105	43, (41)	62, (59)		
Age				0.946	0.331
<50 y	30, (29)	15, (14)	15, (14)		
≥50 y	75, (71)	28, (27)	47, (45)		
Tumor size				5.156	0.023
T1	66, (63)	21, (20)	45, (43)		
T2-T3	39, (37)	22, (21)	17, (16)		
Lymph nodal status				0.750	0.387
N0	65, (62)	24, (23)	41, (39)		
N1-N2	40, (38)	19, (18)	21, (20)		
Tumor stage				7.562	0.023
Stage I	46, (44)	12, (26)	34, (74)		
Stage II	39, (37)	21, (54)	18, (46)		
Stage II	20, (19)	10, (50)	10, (50)		
Grade				0.000	1
G1-G2	83, (79)	34, (32)	49, (47)		
G3	22, (21)	9, (9)	13, (12)		
Ki67				15.853	<0.001
<20%	45, (43)	8, (8)	37, (35)		
≥20%	60, (57)	35, (33)	25, (24)		
Perineural invasion				0.005	0.796
Absent	87, (83)	35, (33)	52, (50)		
Present	18, (17)	8, (8)	10, (10)		
Histological type				4.079	0.131
NST	90, (86)	40, (38)	50, (48)		
Lobular	11, (10)	3, (3)	8, (8)		
Other	4, (4)	0, (0)	4, (4)		
PR				6.108	0.013
<20%	27, (26)	17, (16)	10, (9)		
≥20%	78, (74)	26, (25)	52, (50)		
CT				7.720	0.005
Yes	60, (57)	32, (30)	28, (27)		
No	45, (43)	11, (10)	34, (32)		

No special type, NST; progesterone receptor, PR; chemotherapy, CT.

**Table 3 jcm-11-06987-t003:** Univariate analysis of prognostic variables for Disease Free Survival (DFS) and Breast Cancer Specific Survival (BCSS).

	DFS		BCSS	
	HR (95% CI)	*p*-Value	HR (95%CI)	*p*-Value
Age		0.12		0.082
<50	1		1	
≥50	3.2 (0.74–14)		3.7 (0.85–17)	
PR		0.38		0.46
<20%	1		1	
≥20%	0.64 (0.24–1.7)		0.68 (0.25–1.9)	
c-MET		0.0027		0.0013
Low c-MET <50%	1		1	
High c-MET ≥50%	9.6 (2.2–42)		13 (1.7–99)	
Ki67		0.017		0.021
<20%	1		1	
≥20%	6.1 (1.4–27)		11 (1.4–83)	
Lymph nodal status		0.16		0.28
N0	1		1	
N1-N2	2 (0.77–5.2)		1.7 (0.65–4.6)	
Tumor size		0.0018		0.0024
T1	1		1	
T2-T3	6 (2–18)		7 (2–25)	
Tumor stage		0.0062		0.0058
Stage I-II	1		1	
Stage III	3.9 (1.5–10)		4.3 (0.44–4.3)	
Grade		0.76		0.59
G1-G2	1		1	
G3	1.2 (0.39–3.6)		1.4 (0.44–4.3)	
Perineural invasion		0.081		0.034
Absent	1		1	
Present	2.5 (0.89–7.3)		3.1 (1.1–9.1)	
Histological Type		0.69		1
NST	1		1	
Lobular/Other	1.3 (0.37–4.5)		1 (0.23–4.4)	
CT		0.076		0.17
No	1		1	
Yes	2.8 (0.9–8.5)		2.2 (0.71–6.9)	

## Data Availability

The datasets used analysed during the current study are available from the corresponding author on reasonable request.

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
