# Peer review of "Expression of c-MET in Estrogen Receptor Positive and HER2 Negative Resected Breast Cancer Correlated with a Poor Prognosis"

_jcm, 2022, doi:10.3390/jcm11236987_

Round 1
Reviewer 1 Report
The authors aim to analyze the role of the mesenchymal-epithelial transition factor (c-MET) receptor overexpression in breast cancer patients. They reviewed the clinical features and outcomes of 105 breast cancer patients with estrogen receptor-positive HER2 negative (ER+/HER2-) and find that high c-MET values significantly correlated with tumor size, high Ki67 and low (<20%) progesterone receptor expression. In survival analysis, they also noticed that high c-MET expression correlated with poor survival outcomes. I have several comments for the authors’ consideration to further improve the manuscript.
1. In the methods part, the author mentioned that “We excluded all cases of breast carcinoma in situ, locally advanced and metastatic BC at the time of diagnosis, and patients who received neo-adjuvant chemotherapy.” The author needs to provide more specific information about the exact number of patients that they excluded for each steps.
2. The author could use the TCGA and METABRIC databases to do further analysis of the role of c-MET in breast cancer patients (e.g Survival analysis and clinic information). TCGA is a pan-cancer database that included nearly 33 different types of cancer information (RNA-Seq and clinic data like the clinic stage and survival information). You could use the Cbioportal website to get access to these data.
3. If possible, I suggest the author could add more bioinformatic analysis in this paper, like the enrichment pathway analysis (Hallmark/ GO/ KEGG) in the TCGA database to further explore the role of c-MET in breast cancer patients.
4. There are some minor language errors. The authors should be revised the manuscript with an English language editor to make it more readable.
Author Response
We thank the referee for her/his kind suggestions that help to meliorate our manuscript. Please see below our responses to the concerns.
1) In the methods part, the author mentioned that “We excluded all cases of breast carcinoma in situ, locally advanced and metastatic BC at the time of diagnosis, and patients who received neo-adjuvant chemotherapy.” The author needs to provide more specific information about the exact number of patients that they excluded for each steps.
Answer: We excluded 50 cases of breast carcinoma in situ, 60 locally advanced and 30 metastatic BC at the time of diagnosis, and 30 patients who received neo-adjuvant chemotherapy.
2. The author could use the TCGA and METABRIC databases to do further analysis of the role of c-MET in breast cancer patients (e.g Survival analysis and clinic information). TCGA is a pan-cancer database that included nearly 33 different types of cancer information (RNA-Seq and clinic data like the clinic stage and survival information). You could use the Cbioportal website to get access to these data.
Answer: We have performed the suggested analyses on the publicly available data set and we have reported the correlation between cMet RNA and protein expression and patient survival. We have added these data in the manuscript and appropriately discussed as correctly suggested by the referee. Notably, no available previous information about the protein cMet expression and survival of patients is available and our report is the first one reporting this correlation in breast cancer.
3. If possible, I suggest the author could add more bioinformatic analysis in this paper, like the enrichment pathway analysis (Hallmark/ GO/ KEGG) in the TCGA database to further explore the role of c-MET in breast cancer patients.
Answer: As appropriately suggested by the referee we have performed both GO and KEGG analysis in the TCGA database about the functions of cMET in breast cancer and we have added these data again in the manuscript with related appropriate discussion.
Please find attached the specific answers about the bioinformatic analyses.
4. There are some minor language errors. The authors should be revised the manuscript with an English language editor to make it more readable.
Answer: The manuscript was revised by an English mother tongue for the language style and misreadings.

Reviewer 2 Report
The authors investigate the relationship between c-MET expression levels and prognosis by the retrospective review of the clinical features and outcomes of 105 women with estrogen receptor positive HER2 negative (ER+/HER2-) resected breast cancer.
The topic is very relevant since breast cancer is a widely heterogeneous disease which translate in different clinical approaches and treatment sensitivity. To date the prognostic tests based on gene expression are not widely available for routine clinical practice, therefore therapeutic decisions are still based on traditional parameters such as histological subtype, grading and proliferation index.
The main contribution and strengths of the paper is the demonstration of a statistically significant correlation between c-MET overexpression and tumor size and stage, PR levels and Ki67 index. The results make c-MET a novel prognostic factor to be investigated to validate the clinical relevance and applicability in ER+/HER2- early BC.
The area of weakness is related to the limited number of cases and the lack of a standardized method for immunohistochemical staining and scoring of c-Met expression.
Overall, the topic is covered with completeness and relevance.
Minor comment:
Line 129: format the legend of the figure
Line 206: insert the table on a single page
Author Response
We thank the referee for her/his kind consideration about our manuscript. We are grateful for her/his comments. We clearly agree with her/his consideration about the standardization of the methods. On this view, we have to consider, however, that all the immunohistochemical evaluations were performed in a centralized manner in a Pathology lab with great experience about this kind of methods and that have received also the release for the certification for the diagnostic procedures from the Italian Agency for Drugs and Diagnostics. We are also aware that our series is limited and needs additional work on a larger cohort of patients in order to validate the results. A follow-up study is presently ongoing on this subste of breast cancer patients.
Minor changes.
Minor comment:
1) Line 129: format the legend of the figure.
We have formatted the figure legend
2) Line 206: insert the table on a single page
We have inserted the table on a single page.
Round 2
Reviewer 1 Report
The authors have answered my questions and the paper has been improved.